**Subject Category:**
Biology (whole organism)

biomechanics/behaviour/neuroscience

locomotion, age, falls, balance control, strategy selection

**Author for correspondence:**
M. Pijnappels
e-mail: m.pijnappels@vu.nl

# Does misjudgement in a stepping down paradigm predict falls in an older population?

N. Kluft[1], S. M. Bruijn[1,2], R. H. A. Weijer[1], J. H. van Dieën[1] and M. Pijnappels[1]

[1]Department of Human Movement Sciences, Vrije Universiteit Amsterdam, Amsterdam Movement Sciences, Amsterdam, The Netherlands
[2]Institute Brain and Behaviour Amsterdam, Amsterdam, The Netherlands

(iD) NK, 0000-0001-6504-8724

Although measures of actual and perceived physical ability appear to predict falls in older adults, a disparity between these two, also known as misjudgement, may even better explain why some older adults fall, while their peers with similar abilities do not. Therefore, we investigated whether adding a misjudgement term improved prediction of future falls. Besides conventional measures of actual (physical measures) and perceived abilities (questionnaires), we used a stepping down paradigm to quantify behavioural misjudgement. In a sample of 55 older adults (mean age 74.5 (s.d. = 6.6) years, 33 females and 20 fallers over a 10-month follow-up period), we tested the added value of a misjudgement term and of a stepping-down task by comparing experimental Bayesian logistic-regression models to a default null model, which was composed of the conventional measures: Falls Efficacy Scale international and QuickScreen. Our results showed that the default null model fitted the data most accurately; however, the accuracy of all models was low (area under the receiver operating characteristic curve (ROC) $\leq 0.65$). This indicates that neither a misjudgement term based on conventional measures, nor on behavioural measures improved the prediction of future falls in older adults (Bayes Factor$_{10} \leq 0.5$).

## 1. Background

Manoeuvring safely through the environment requires the ability to perceive the biomechanical requirements of an encountered task, and the ability to judge whether these requirements lie within one's physical ability. This entails that safe behaviour depends not only on an accurate perception of the task at hand, but also on a

precise perception of one's physical ability [1]. An error in either of these perceptions could lead to misjudgement, by either over- or underestimation, which may result in suboptimal movement strategy selection. As ageing is accompanied by a decline in cognitive and physical abilities, inaccurate judgement may become more apparent at an older age. One consequence of our ageing society is the increase of falls and fall-related injuries [2–4], which may (partly) be explained by such misjudgements [5].

Many of the existing models predicting falls in older adults already incorporate either a measure of actual ability or a measure of perceived ability. For instance, the Timed Up and Go test [6], Berg balance scale [7], Physiological Profile Assessment [8] or QuickScreen [9] are all attempts to predict falls by assessing the physical state of the older individual. On the other hand, the gait- and falls-efficacy scales [10,11] are measures that assess the individual's confidence to do certain activities, which closely resembles one's perceived ability. However, although previously suggested [12], a misjudgement term by means of an interaction between the perceived and actual ability has not yet been incorporated in fall prediction models.

Misjudgement between older adults' perceived and actual ability has been studied in stair climbing [13], stepping over obstacles [1,5,14], balance and reaching ability [15,16] and gait [17,18]. Although each of these tasks requires adequate balance control, the use of these tasks for determining misjudgement may be hampered by two concerns. First, the perceived ability measure is commonly determined by asking participants explicitly for their maximum ability, which could promote socially desirable responses [19]. Second, the perceived ability was measured in a static posture and lack of optic flow might hamper the perception of task requirements [20,21].

Recently, attempts have been made to resolve these concerns. Butler and co-workers [17] measured the perceived ability more implicitly by designing a dynamic task in which healthy older participants were exposed to a walking track with six different walkways varying in levels of difficulty. These levels were presented so that the easiest option was presented farthest away. Participants were instructed to complete the track as fast as possible, thereby selecting the walkway that they believed would be optimal in terms of both safety and efficiency. However, participants were not allowed to actually walk across the walking track, and the misjudgement was determined by predicting their performance on the selected track on the basis of the gait characteristics during another walking bout. Interestingly, they found that overestimation was associated with a higher incidence of future falls. Similarly, Kluft and co-workers [1] created a tapered paper 'river', which participants needed to cross as quickly as possible, inducing a trade-off between efficiency (i.e. be as quick as possible) and safety (i.e. choosing a crossing point that lies within one's stepping ability). Instead of relating the outcome measures to falls, the objective was to investigate whether a similar degree of misjudgement was observed between multiple stepping tasks involving explicit and implicit estimates of self-perceived ability. In both studies, the instructions were to perform the task as fast as possible. However, as walking speed changes the biomechanical requirement of the task at hand, the selected walking speed should be incorporated in the judgement people make, and should therefore be considered when assessing misjudgement in dynamic gait tasks.

We propose a paradigm that alleviates these concerns, namely stepping down to a lower level, while maintaining a constant walking speed. When stepping down, humans select a strategy and perform either a heel or a toe landing [22,23]. The latter strategy is considered to be the 'more stable but more [physically] demanding' strategy [24, p. 345], and is commonly selected with larger height differences. Compared to a toe landing, a heel landing is considered to be less physically demanding, but balance control becomes more challenging, as less kinetic energy is absorbed in landing at the lower level [23,25]. A distinct height at which an individual switches the preferred strategy, the critical switching height ($h_{crit}$) appeared highly variable between participants [23,24]. This behavioural measure $h_{crit}$ can be regarded as a measure of perceived ability, but does not rely on any instructions regarding the strategy selection. Another advantage of the stepping-down paradigm, is that participants are less aware of the strategy selection they make during this task. Moreover, the judgement is dynamic as the participants are moving towards the height difference.

In this study, we used the stepping-down paradigm to obtain a behavioural measure of misjudgement. To estimate misjudgement, we need to relate one's perceived ability to one's actual physical ability (see [23] for a direct comparison). For the current study, we defined the actual physical ability as a composite score of the stepping performance on two stepping tasks. The first stepping task quantified the participant's maximal step length [1], whereas the second task quantified the participant's ability to step over an obstacle [5].

Our main objectives are to understand (1) whether adding a misjudgement term can help to predict fall risk and (2) whether these behavioural measures improve the identification of prospective fallers compared to conventional physical and cognitive measures. We predict that overestimation is

associated with prospective falls, and thus a combination of high perceived and low actual ability should be associated with higher fall incidence. Hence, an interaction between perceived ability (i.e. $h_{crit}$) and actual ability, as suggested by Weijer *et al.* [12], would improve the power to predict fall risk compared to a model based on merely physical and falls efficacy measures.

# 2. Methods

## 2.1. Participants

Sixty-two older adults (age ≥ 65 years, median age 73.5 [IQR 10] years, 37 females, 20 fallers) participated in the study. Participants were included when they had a mini-mental state examination score above 18 points (i.e. no to mild cognitive decline, excluding severe cognitive impairment), were able to walk 20 m continuously without becoming short of breath, experiencing dizziness, or perceiving pain in or pressure on the chest. The complete experimental procedure was explained prior to any experimentation, and all participants provided written consent. This study was part of a larger longitudinal study on self-perception, gait quality and physical activity (Veilig In Beweging Blijven, [12]). The Vrije Universiteit's ethics committee (i.e. Vaste Commissie Wetenschap en Ethiek) approved all experimental procedures (# VCWE 2016-147).

## 2.2. Protocol

Participants were invited for the assessment once at the Vrije Universiteit. After the assessment, they were asked to keep a fall diary during a follow-up period of 10 months.

## 2.3. Falls

Falls were monitored for 10 months using fall diaries and monthly telephone calls to ask about the occurrence of falls over the previous month. The baseline measurement was scheduled in the first month that falls were monitored. A fall was defined as 'inadvertently coming to rest on the ground, floor or other lower level, excluding intentional change in position to rest on furniture, wall or other objects' [26, p. 1]. Participants were classified as fallers when one or more falls were recorded. A total of 55 out of 61 participants completed the entire follow-up.

## 2.4. Assessments

### 2.4.1. Actual physical ability (composite score).

During the assessment, participants performed a set of motor tasks from which a composite physical ability score ($x_{act}$) was calculated. This composite score was calculated on the basis of a larger dataset from a cohort study of our group [12], and consisted of a weighted average of the participant's maximal step length (Step length) and maximum performance to step over (Step over) an obstacle. The weights in the equation (see equation (2.1)) were determined by the first principal component, which explained 90% of the variance.

$$x_{act} = 0.405967 \cdot \left( \frac{\text{Step over}}{\text{Leg length}} \right) + 0.91389 \cdot \left( \frac{\text{Step length}}{\text{Leg length}} \right). \tag{2.1}$$

### 2.4.2. Perceived ability (stepping-down protocol).

Two equally sized platforms (1.2 × 2 m) were placed in front of each other (figure 1). Six stair-like wooden blocks supported one of the platforms at variable heights. The height difference between the platforms could be adjusted between 2.5 and 15 cm, in steps of 2.5 cm. At the far end of the second platform, a 28.5 by 46.5 cm target was fixed on the surface of the platform. Parallel to the two platforms, a light emitting diode (LED) strip, with 30 LEDs per metre, was placed at eye height. A small beam of 20 consecutive LEDs moved along the light strip at a speed of 1.1 m s$^{-1}$.

Participants were instructed to walk from the starting point of the first platform towards the end point on the second platform, thereby stepping down at the height difference, while walking at the same speed

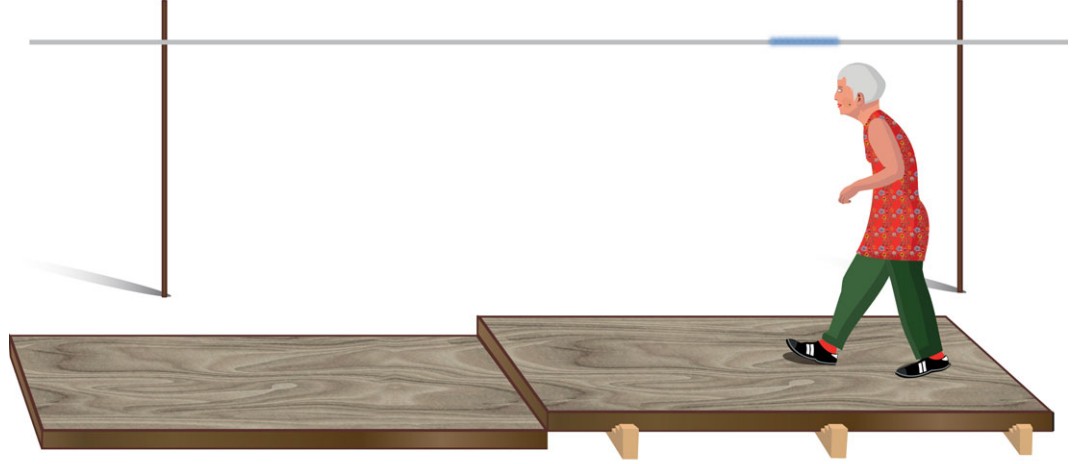

**Figure 1.** Illustration of the experimental set-up. Participants were asked to step down at the end of the first platform. The step height was adjustable and ranged from 0.025 to 0.15 m. The participant was instructed to adopt a walking speed of 1.1 m s$^{-1}$, as indicated by an LED strip alongside the platform.

as the moving LEDs. Upon arrival at the target, participants stood still for a moment and returned to the starting point. The initial height difference was 5 cm. For each height difference, the procedure was repeated six times. Based on the strategy selection during these six repetitions, the next height was determined using a fixed protocol (described in [23]). In brief, if one of the six repetitions resulted in a heel landing, the height difference was increased by 5 cm. We continued this approach of increases in height differences, until all six repetitions at a given height were toe landings, or the maximum height of 15 cm was reached. Finally, the height difference was set at a level of 2.5 cm higher than the height at which only heel landings occurred. This fixed protocol ensured that all heights between the height at which only toe landings and the height at which only heel landings occurred were observed, within a reasonable timeframe.

## 2.5. Data collection and analysis

During the assessment, the stepping-down strategies of the participant were classified based on visual inspection, to adjust the height difference between the platforms using the protocol described above. However, for objective analyses of the strategies, all trials were captured using two video cameras and were categorized *post hoc* by two independent pairs of raters. Conflicting categorizations were rated once more by the first author.

A nominal logistic regression model was fitted to the landing-strategy data of each participant. This resulted in a sigmoid model that described the landing strategy as a function of height difference. The height at which the probability of a toe landing ($P_{toe}$) equalled the probability of a heel landing ($P_{heel}$) was defined as $h_{crit}$ [23] normalized to participant's leg length.

## 2.6. Statistical procedures

The inter-rater agreement of the categorized stepping-down strategies was determined by assessing Krippendorff's alpha ($K\alpha$) (see [27] for a discussion on reliability measures). A $K\alpha$ near one indicates excellent reliability, while a value of zero is indicative for no reliability, and a negative $K\alpha$ indicates a systematic disagreement.

To discriminate the people who fell from those who did not fall in the following months, four Bayesian logistic regression models were constructed (equations (2.2)–(2.5)), including actual physical ability ($x_{act}$) and perceived ability (i.e. $h_{crit}$) terms. Bayesian inferences enhance the interpretation of the model, as posterior distributions of the model parameters can be computed, thereby incorporating the uncertainty in the parameters. Moreover, the evidence in favour of a model can be quantified by deploying Bayes factors (BF), which facilitates comparisons of models. Experimental models ((2.3)–(2.5)) were compared to a default null model, based on two input variables commonly accepted in fall prediction and indicative for actual and perceived fall risk: (i) the QuickScreen (QS, [9]) and

(ii) the Dutch version of the falls-efficacy scale international (FESi, [11,28]), respectively. The probability of a fall ($P_{null}$) as a function of the FESi and the QS in the first model is given by the equation:

$$P_{null}(FESi, QS) = \frac{1}{1 + e^{-(\beta_0 + \beta_1 \cdot FESi + \beta_2 \cdot QS)}}. \tag{2.2}$$

Here and in the rest of the manuscript, the $\beta_i$ are predictor coefficients that are obtained by fitting the logistic regression to the prospective fall data. The QuickScreen identifies risk factors by assessing balance, gait, strength and endurance to examine lower-limb function, and a composite score of the risk factors is associated with falls [9]. The FESi, a self-report questionnaire, examines one's concern about falling in a variety of gait-related daily activities [11], and can reliably distinguish between fallers and non-fallers [29].

### 2.6.1. First model

The first experimental model contained an updated version of the null model. Delbaere et al. [30] showed that the discrepancy between older adults' perceived and physiological fall risk can help to explain falls. This discrepancy in fall risk is implemented by adding an interaction term (i.e. FESi × QuickScreen) to the null model. Although this model incorporates a misjudgement term, the term is not directly linked to participants' actual movement behaviour. In the first experimental model, the probability of a fall ($P_1$) as a function of the FESi and the QS is given by the equation:

$$P_1(FESi, QS) = \frac{1}{1 + e^{-(\beta_0 + \beta_1 \cdot FESi + \beta_2 \cdot QS + \beta_3 \cdot FESi \times QS)}}. \tag{2.3}$$

### 2.6.2. Second model

The second experimental model (equation (2.4)) contained the $x_{act}$ and the height chosen for switching at the stepping down paradigm ($h_{crit}$). In the second experimental model, the probability of a fall ($P_2$) as a function of $x_{act}$ and $h_{crit}$ is computed by the equation:

$$P_2(x_{act}, h_{crit}) = \frac{1}{1 + e^{-(\beta_0 + \beta_1 \cdot x_{act} + \beta_2 \cdot h_{crit})}}. \tag{2.4}$$

This model incorporated participants' physical ability and perceived ability, represented by the $x_{act}$ and the $h_{crit}$ terms, respectively. Although this model contained information about the physical and the perceived ability, it lacks a misjudgement term.

### 2.6.3. Third model

The next model incorporated the misjudgement term. Since the misjudgement can be treated as a weighted form of the perceived ability ($h_{crit}$), it should be related to one's actual physical ability ($x_{act}$). As the first model, the third experimental model (equation (2.5)) was extended with an interaction of the $x_{act}$ and $h_{crit}$ variables. The probability of a fall ($P_3$) in the third model is given by the equation:

$$P_3(x_{act}, h_{crit}) = \frac{1}{1 + e^{-(\beta_0 + \beta_1 \cdot x_{act} + \beta_2 \cdot h_{crit} + \beta_3 \cdot x_{act} \times h_{crit})}}, \tag{2.5}$$

in which the interaction $x_{act} \times h_{crit}$ represents the misjudgement term.

### 2.6.4. Input variables and priors

For all models, each input variable was standardized, and weakly informative priors were assigned to all model predictors [31]. The prior for the model's intercept followed a zero-centred Cauchy distribution with scale 10, and a zero-centred Cauchy distribution with scale 2.5 was selected for the predictor coefficients priors, thereby following the recommendations of [32]. Collinearity of the predictors was assessed using the variance inflation factor (VIF), and showed that there was no severe collinearity that would have affected the analysis (VIF <1.9). Hence, correlations between predictors were low ($r_{(Quickscreen, FESi)} = 0.06$; $r_{(x_{act}, h_{crit})} = 0.20$).

### 2.6.5. Model outcome measures

For each of our experimental models, the median of the $\beta_i$ coefficient posterior distribution and the corresponding 95% highest density interval (95% HDI, Bayesian analogue to the 95% confidence

**Table 1.** Participant descriptives, response and explanatory variables. When the variable was normally distributed the mean M with standard deviation (s.d.) were reported, otherwise the median Mdn with interquartile range [IQR] were reported. Solely for count data the number and percentage $n$(%) of the total sample size were reported.

| descriptives ($n = 55$ older adults) | | | | |
|---|---|---|---|---|
| *descriptive variables* | | | | |
| age | 74.5 | (6.6) | M(s.d.) | years |
| females | 33 | (60%) | $n$(%) | persons |
| unique medication | 2 | [4] | Mdn[IQR] | units/week |
| falls in the past | 1 | [1.5] | Mdn[IQR] | falls |
| MMSE | 28 | [2.5] | Mdn[IQR] | points |
| body weight | 74.3 | (12.3) | M(s.d.) | kg |
| body height | 168.9 | (8.8) | M(s.d.) | cm |
| grip strength | 27.5 | [7.4] | Mdn[IQR] | kg |
| Max. knee extension moment | 7.4 | [3.2] | Mdn[IQR] | N m |
| *response variable* | | | | |
| fallers[a] | 20 | (36.4%) | $n$(%) | persons |
| *explanatory variables* | | | | |
| FESi | 20 | [5] | Mdn[IQR] | points |
| quickScreen (QS) | 4 | [2] | Mdn[IQR] | risk factors |
| $x_{act}$ | 1.23 | (0.21) | M(s.d.) | (arbitrary variable) |
| $h_{crit}$ | 7.1 | [5.7] | Mdn[IQR] | cm |

[a]Considered a faller when 1 or more falls occurred.

interval) were calculated. Subsequently, we computed the leave-one-out cross-validation (LOO; [33]), quantifying the model's out-of-sample prediction accuracy, and the Watanabe–Akaike information criterion (WAIC, [34]), which is a Bayesian equivalent of the Akaike information criterion. Then, each experimental model was compared with the null model, and the evidence in favour of the experimental model relative to the null model was quantified using Bayes factors [35]. Analyses were performed using custom-made software in Matlab R2018a, and all Bayesian analyses were performed using the 'brms' package [36] in R [37].

# 3. Results

## 3.1. Participant descriptives

Data of one participant were excluded because this participant was unable to complete the stepping-down protocol. Furthermore, six participants did not complete the 10 months follow-up. Twenty (36.4%) of the remaining 55 older participants were considered fallers (falls $\geq 1$, see table 1 for participant descriptives).

## 3.2. Strategy rating agreement

The two independent pairs of raters disagreed on only six stepping-down strategies (0.6% of all trials), suggesting a very high inter-rater reliability, which was confirmed by the high Krippendorff's alpha value ($K\alpha = 0.99$). The six conflicting trials were classified once more by one of the authors. On average participants switched strategies ($h_{crit}$) at a height of 8.0 cm (s.d. = 4.8 cm).

## 3.3. Fall prediction models and contribution of misjudgement

The models coefficients and their statistics are presented in table 2. The QuickScreen's 95% HDI in the null and first model excluded zero, even as the intercept in all models. However, for all other coefficients, the

**Table 2.** Posterior summaries of logistic regression coefficients. The median, the standard error, the lower and upper boundary of the 95% highest density interval (95%HDI), and the Bayes factor BF$_{10}$ of the coefficient's posterior distribution are depicted in this table.

| model | equation | coefficient | median | std. error | 95%HDI | | BF$_{10}$ |
| --- | --- | --- | --- | --- | --- | --- | --- |
| | | | | | lower | upper | |
| null model | (2.2) | | | | | | |
| | | Intercept | −0.60 | 0.29 | [−1.24 | −0.04] | 0.20 |
| | | FESi | 0.35 | 0.53 | [−077 | 1.43] | 0.22 |
| | | QuickScreen | 1.02 | 0.61 | [−0.17 | 2.22] | 0.84 |
| first model | (2.3) | | | | | | |
| | | Intercept | −0.63 | 0.30 | [−1.21 | −0.06] | 0.23 |
| | | FESi | 0.42 | 0.56 | [−0.81 | 1.60] | 0.24 |
| | | QuickScreen | 1.03 | 0.59 | [−0.19 | 2.27] | 0.86 |
| | | FESi × QuickScreen | 0.73 | 1.22 | [−1.70 | 3.39] | 0.44 |
| second model | (2.4) | | | | | | |
| | | Intercept | −0.56 | 0.29 | [−1.12 | 0.00] | 0.15 |
| | | $x_{act}$ | 0.03 | 0.55 | [−1.14 | 1.13] | 0.31 |
| | | $h_{crit}$ | 0.24 | 0.56 | [−0.78 | 1.39] | 0.18 |
| third model | (2.5) | | | | | | |
| | | Intercept | −0.49 | 0.30 | [−1.05 | 0.13] | 0.09 |
| | | $x_{act}$ | 0.23 | 0.63 | [−1.04 | 1.44] | 0.21 |
| | | $h_{crit}$ | 0.25 | 0.61 | [−0.99 | 1.43] | 0.21 |
| | | $x_{act} \times h_{crit}$ | −2.26 | 1.35 | [−5.23 | 0.31] | 2.12 |

**Table 3.** Goodness-of-fit measures. The measures reported here: the leave-one-out cross validation (LOO), Watanabe–Akaike information criterion (WAIC) and the Bayes factor (BF$_{10}$; the evidence of the experimental model relative to the null model). In general, for model comparisons, the smaller the LOO or the WAIC the better the model fits the data.

| model | input variables | equation[†] | LOO | WAIC | BF$_{10}$ |
| --- | --- | --- | --- | --- | --- |
| null model | (FESi, QS) | (2.2) | 74.52 | 74.45 | |
| first model | (FESi, QS, FESi × QS) | (2.3) | 76.14 | 75.89 | 0.44 |
| second model | ($x_{act}$, $h_{crit}$) | (2.4) | 77.40 | 77.33 | 0.27 |
| third model | ($x_{act}$, $h_{crit}$, $x_{act} \times h_{crit}$) | (2.5) | 75.39 | 75.17 | 0.58 |

[†]Reference to equation.

95%HDI spanned zero. The misjudgement terms—the interaction term in the first (FESi × QuickScreen) and third ($x_{act} \times h_{crit}$) model—did not contribute to explaining the data, with coefficient estimates (i.e. median of the posterior parameter distribution) of $\beta_{FESi \times QuickScreen} = 0.73$ and $\beta_{x_{act} \times h_{crit}} = -2.26$, and the Bayes factor for these predictors reaching a value of BF$_{10} = 0.44$ and BF$_{10} = 2.12$, respectively.

The values for the goodness of fit of the models are displayed in table 3. Both measures of the goodness of fit (i.e. LOO and WAIC) favoured the null model, as the lowest values were found for this model. However, the coefficients' BF$_{10}$ were lower than 1 (see BF$_{10}$s of null model in table 2), indicating that FESi and QuickScreen only increased the model's complexity, while their contribution was limited. The BF$_{10}$ values for the experimental models (see BF$_{10}$s for models 1, 2 and 3 in table 3) containing our stepping-down measures were lower than 1, showing that the data support the null model over the other experimental models.

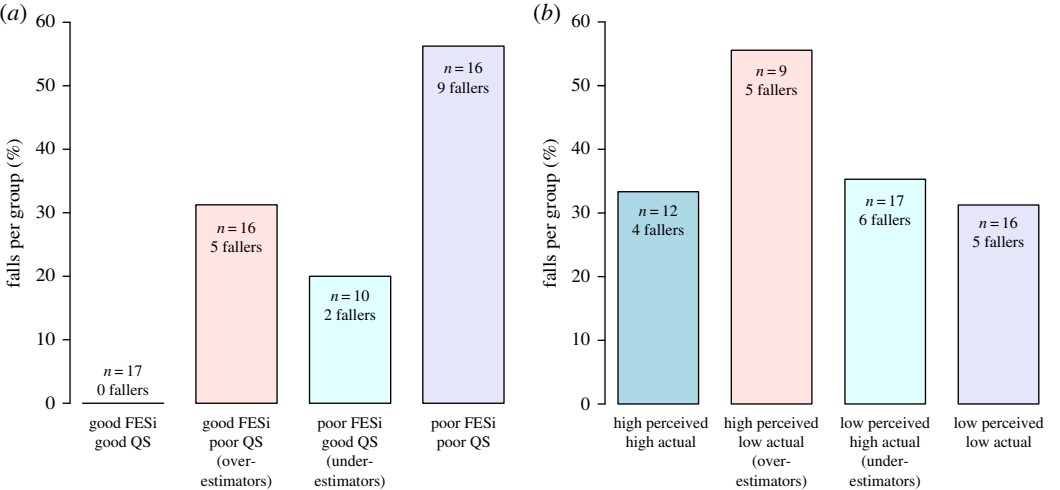

**Figure 2.** The percentage of fallers in four subgroups of the data. (*a*) Groups were separated on the basis of a mean split of the two input variables FESi and QuickScreen (QS). (*b*) The groups were separated on the basis of a mean split of the two input variables $x_{act}$ and $h_{crit}$. The numbers of participants within a group is given by *n*.

## 3.4. Exploratory analysis 1

To gain more insights into our results regarding the interaction, we categorized the participants in four groups based on the actual physical ability and the perceived ability (switching height), and QuickScreen and FESi. Figure 2*a*,*b* shows fall incidence over 10 months for each of the subgroups. The group divisions were made on the basis of (1) a mean split of $x_{act}$ and $h_{crit}$ and (2) a mean split of QuickScreen and FESi. According to this analysis, most participants aligned their movement behaviour with the actual ability measure. The fall incidence on the basis of behavioural measures was highest in the high perceived ability and low actual ability group (fall incidence = 55%), and lowest in the group that had a relative low-perceived and low-actual ability (fall incidence = 31%). In comparison, the fall incidence on the basis of the conventional measures was highest in the poor QuickScreen and poor FESi group (fall incidence = 56%), whereas no one fell in the good QuickScreen and good FESi group (fall incidence = 0%).

## 3.5. Exploratory analysis 2

To better understand and compare the predictive value of the different models, we computed the receiver operating characteristic (ROC) curvature for each model (figure 3). In the ROC curves, the sensitivity (i.e. the proportion of actual positives that are correctly identified) is plotted as a function of the specificity (i.e. the proportion of actual negatives that are correctly identified) for different criterion values (i.e. cut-points) of the model. The area under the ROC curvatures ($ROC_{auc}$) is a measure of how accurately the model can distinguish between fallers and non-fallers. In our sample, the combined model of FESi and QuickScreen reached an accuracy of 0.63 (null model $ROC_{auc}$). The addition of the FESi × QuickScreen interaction did not increase the model's accuracy (first model $ROC_{auc} = 0.65$). The models containing the behavioural measures did not perform better than the conventional measures (second model $ROC_{auc} = 0.57$, and third model $ROC_{auc} = 0.62$).

# 4. Discussion

The present study was designed to investigate the use of a stepping-down task as an objective measure of misjudgement and investigated whether adding a misjudgement term can improve fall prediction models. None of the experimental models performed better than the null model that contained solely a clinical measure and questionnaire (i.e. FESi and QuickScreen). Adding a misjudgement term to the null model did not improve the predictive quality of the model (e.g. *null model's* LOO = 74.52, versus *first model's* LOO = 76.14). Moreover, we found that the addition of a misjudgement term, led to only a very minor improvement in prediction of prospective falls when expressed as the interaction between perceived and actual ability of behavioural measures (i.e. $x_{act} \times h_{crit}$) (e.g. *third model's* LOO = 75.39, versus *second model's* LOO = 77.40).

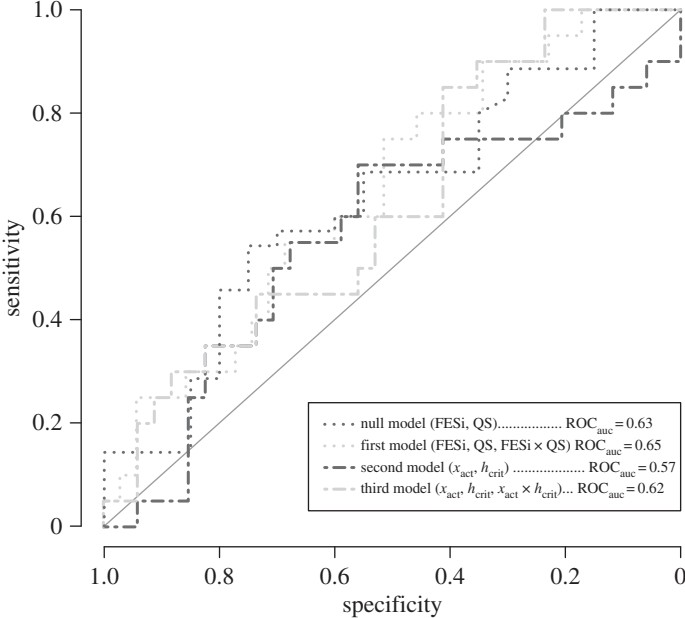

**Figure 3.** Receiver operating characteristic curves for the individual models: sensitivity (the proportion of actual positives that are correctly identified), and specificity (the proportion of actual negatives that are correctly identified), are depicted for different criterion values of the model. The area under the ROC curve ($ROC_{auc}$) is given as a measure of the overall accuracy of the model, a larger $ROC_{auc}$ indicates better performance of the model to distinguish between fallers and non-fallers. The diagonal line (with an $ROC_{auc}$ of 0.5) represents a non-discriminatory model.

The misjudgement predictor (i.e. $\beta_{x_{act} \times h_{crit}}$) estimate was negative, suggesting that participants were less likely to have fallen in the next months when the actual ability corresponded with the perceived ability. Previous studies showed that particularly overestimation of one's ability was associated with fall risk [5,17]. However, we found that a model containing the misjudgement predictor was only 2.12 times more likely than a model containing solely $h_{crit}$ and $x_{act}$, meaning that the data were indecisive in supporting one model above the other (i.e. $BF_{10} = 2.12$). Hence, we can conclude that if there is an association of misjudgement with falls, this association is weak.

Conventional clinical fall prediction models provide only poor to fair predictive ability [38–40]. Our null model comprised two of those conventional measures: the QuickScreen and the FESi. Both measures have demonstrated reasonable accuracy, in terms of the area under the receiver operator curve ($ROC_{auc}$), to predict falls (QuickScreen $ROC_{auc} = 0.72$ [9]; FESi $ROC_{auc} = 0.67$ [29]). However, we found a lower accuracy ($ROC_{auc} = 0.63$) after combining these measures, while our sample was comparable with previous reports (cf. [9,29]). To illustrate this, our sample consisted of older individuals with levels of concern about falling of $22.3 \pm 5.3$ points on the FESi (ranging from a score of 16 to 43), which is somewhat low in comparison with a sample mean of 28 in a Dutch population [28], but well within the range that was reported to be associated with prospective falls (e.g. mean of $22.1 \pm 6.4$ in [30]). Moreover, the median number of risk factors identified by the QuickScreen in our sample was 4, which was higher than what was identified in an external validation study (median number of risk factors = 3 [41]). Furthermore, it should be noted that the QuickScreen tool's $ROC_{auc}$ values were developed in a larger cohort and on the basis of multiple falls data. Recurrent fallers are likely to suffer from more chronic impairments in either perceived or actual ability. Only 14 participants in our study fell more than once during follow-up, which was too small a number to distinguish recurrent fallers from occasional fallers and non-fallers.

In this study, we computed the misjudgement by relating $h_{crit}$ to the composite score $x_{act}$. These measures were used to reflect the participant's perceived and actual physical ability, but the measures are not directly related as they are assessed with separate tasks (cf. [1,5,18]). Measures that are more directly related would give a better indication of when a person truly exceeds his or her actual physical ability. In an earlier approach to quantify the misjudgement, we determined the actual physical ability during unexpected stepping down. The reactive behaviour that was provoked by the unexpected stepping down was quantified, and the ability to absorb kinetic energy reflected the actual physical ability. Hence, this gives a more direct quantification of the misjudgement. However,

this approach can only be applied in fit older adults and would not be feasible in frailer older adults. To our knowledge, there is no test available that directly links the perceived and actual physical ability, without priming or biasing the participant on his/her behaviour.

Participants' perceived ability was assessed using a stepping-down paradigm. The strength of such a paradigm is that the perceived ability is not explicitly assessed. However, the stepping-down behaviour might have been affected by the participant's anthropometry. For instance, limited dorsiflexion range-of-motion could induce toe-off earlier in the gait cycle, which may affect one's stability. This physical limitation may drive participants to select a toe-landing strategy where a heel landing would have been more appropriate. Hence, we recommend to take passive/active ankle range of motion into account in future research that implements a stepping-down paradigm.

We investigated whether misjudgement in an experimental setting was associated with higher fall risk in daily life. Therefore, we assumed that the movement selection in daily life can be derived by assessing movement selection in an experimental setting. The transfer of the misjudgement between several (explicit) stepping tasks seems to be weak [1], which suggests that the misjudgement is highly task specific. Yet, how the misjudgement measure relates to daily life movement selection has not been studied yet. Furthermore, we assumed that an inappropriate strategy selection in daily life would lead to falls. However, a fall is a manifestation of an impaired system, a system that—even if impaired—has a very low error rate (a fall rarely occurs). Although falls were recorded prospectively over 10 months, the number of falls might not be a perfect measure to quantify the impairment of a system in daily life.

In this study, a Bayesian statistical analysis was performed, in contrast to the conventional frequentist statistics as generally used in falls prediction. Accidental falls in older adults form a noisy measure, and the effect sizes are often small and unlikely to be known prior to the execution of the study. A meaningful power analysis is therefore hard to perform at the start of an experiment. Unlike conventional frequentist statistics, in a Bayesian framework, one can perform a study in which samples are added until a certain level of evidence has been reached without making inferential errors [42]. As falls in older adults are generally monitored on a yearly basis by medical bodies, studies performing statistics in a Bayesian framework can improve the accuracy of the model parameters by incorporating information of the preceding year. Therefore, it is advisable to consider performing Bayesian statistical analysis in fall-related research.

# 5. Conclusion

Our findings showed that a misjudgement term does not improve the prediction of future falls in older adults, be it an interaction term between conventional measures or between behavioural measures of perceived and actual physical ability.

Ethics. All participants signed informed consent, and all procedures were approved by the local ethics committee (# VCWE 2016-147).

Data accessibility. The datasets analysed for this study and all figures can be found in an Open Science Framework repository (https://osf.io/5erjw/).

Authors' contributions. All authors contributed to the conception and design of the study. N.K. built the set-up, and N.K. and R.H.A.W. collected the experimental data. N.K. developed the code for the data analysis. N.K. took the lead in writing the manuscript and designed the figures. All authors provided critical feedback and helped shaping the research, analysis and manuscript. M.P. supervised the project. All authors contributed to manuscript revision, read and approved the submitted version.

Competing interests. No conflicts of interest are declared by the authors.

Funding. This work was supported by a VIDIgrant (grant no. 91714344) from the Dutch Organization for Scientific Research (NWO). S.M.B. was funded by a VIDI grant (grant no. 016.Vidi.178.014) from the Dutch Organization for Scientific Research (NWO).

Acknowledgements. The authors thank Jorrit Cornelissen, Lauren van Etten, Richella Hens, Lian van Rijn, Daphne Suiker, Mark Melman and Martine Rog for their assistance during data collection. We wish to express our gratitude to Leon Schutte and Siro Otten for the development of the experimental set-up.

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
