## [Reviewer comments · Royal Society Open Science]

Review History

RSOS-190786.R0 (Original submission)

Review form: Reviewer 1

Is the manuscript scientifically sound in its present form?

Yes

Are the interpretations and conclusions justified by the results?

Yes

Is the language acceptable?

Yes

Is it clear how to access all supporting data?

Yes

Do you have any ethical concerns with this paper?

No

Have you any concerns about statistical analyses in this paper?

I do not feel qualified to assess the statistics

Recommendation?

Accept with minor revision (please list in comments)

Comments to the Author(s)

This study examined whether adding a misjudgement term improved prediction of future falls compared with a) conventional measures of actual (physical measures) and perceived abilities (questionnaires), and b) a novel stepping down paradigm that could quantify behavioural misjudgement. The sample comprised 55 older adults of which 20 reported 1+ falls over a 10 month follow-up period. The main findings were that a default model comprising the conventional measures (FESI and QuickScreen) fitted the data most accurately and that the inclusion of misjudgement terms did not improve the prediction of future falls in older adults. It was also found the accuracy of all models was low (area under the receiver operating characteristic curve (ROC) ≤ 0.64).

This is an interesting study that uses novel approaches to provide greater understanding of fall risk in older people. The complementary use of both conventional and behavioural measures of fall risk and misjudgement is a major strength of the paper. I have the following issues for consideration.

It is indicated in the methods that the exact measure of physical ability was derived from previous research. Was this measure validated with respect to falls in this larger study?

The combination of the small sample for a fall risk study, a relatively healthy sample and the choice of QuickScreen may have limited the study to detect differences between the faller groups. As indicated in the discussion, real world falls is a "noisy" outcome measure and larger samples ($N > 100$) are usually used to partially address this issue. Studies often focus on multiple falls as this group appear to have chronic physical impairments, whereas those who suffer an occasional fall often perform as well as non-fallers in physical assessments. Indeed, ROC characteristics of the QuickScreen tool (as used here) were based on an outcome of multiple (2+) fallers. The small QuickScreen IQR (2) also indicated little fall risk variation in the sample. Given the small sample, it would be problematic to conduct further analyses, but it would be informative to provide the number of participants who fell one time only and to discuss these issues in a limitations section of the discussion.

Please provide an additional panel to figure 3 depicting falls per group in participants categorised by fall risk matches and mismatches as derived from the conventional measures (FESI and QuickScreen).

It would also be instructive to see correlations between the QuickScreen, FESI, exact and hcrit measures.

Review form: Reviewer 2 (Matthieu Boisgontier)**Is the manuscript scientifically sound in its present form?**

Yes

Are the interpretations and conclusions justified by the results?

Yes

Is the language acceptable?

Yes

Is it clear how to access all supporting data?

Yes

Do you have any ethical concerns with this paper?

No

Have you any concerns about statistical analyses in this paper?

No

Recommendation?

Major revision is needed (please make suggestions in comments)

Comments to the Author(s)

Title: Does misjudgment in a stepping down paradigm predict falls in an older population?

Journal: Royal Society Open Science

Manuscript ID: RSOS-190786

Date: July 1, 2019

This manuscript reports a single study designed to test whether misjudged physical abilities contribute to predict falls beyond and above actual and perceived physical abilities. Actual abilities were assessed using the weighted average of maximal step length and maximum performance to step over an obstacle. Perceived abilities were assessed based on a nominal logistic regression model that fitted the toe versus heel landing-strategy data in a 6-height stepping-down protocol. Misjudged abilities were modeled by the interaction term between the actual and perceived abilities. The data from 55 older adults (20 fallers) were included in Bayesian logistic-regression models. Results showed that the interaction terms improved the fit of the model but that the null model was the most accurate one. All models showed low accuracy with areas under the ROC curves $<.66$).

General comments

This manuscript is very well written, it was a pleasure to read it. The introduction provides an accurate and complete overview of the existing literature related to falls and misjudged physical abilities. The approach used to address the research question is clever. The statistics used are appropriate and go beyond the usual standards in our field. The results do not support the hypothesis but provide useful and clear information that will serve the implementation of future study designs.

Specific comments

-Page 5 (Objectives): "We predict that overestimation is associated with prospective falls, and thus a combination of high perceived and low actual ability should be associated with higher fall incidence. Hence, an interaction between perceived ability (i.e., h_{crit}) and actual ability, as suggested by [12], would improve the power to predict fall risk compared to a model based on merely physical and falls efficacy measures."

Clever approach to the research question.

-Page 5 (a) Participants: "Participants were included when they had a mini-mental state examination score above 18 points"

It may be worth making the meaning of this 18-point threshold explicit to the reader (e.g. no to mild cognitive impairment).

-Page 5 (c) Falls: "Falls were monitored for 10 months using telephone calls"

If the frequency (even a rough one) of these phone calls could be reported, this may be useful information to readers.

-Page 6 (ii) Perceived ability (stepping-down protocol):

In future studies, as this task is central to the research question, it may be a good idea to control for passive and/or active ankle range of motion, especially dorsiflexion. Limited range of dorsiflexion forces the heel of the upper foot to leave the step earlier thereby reducing stability and increasing the likelihood that the participants try to contact the lower step as fast as possible with the lower foot, which would be done by contacting the step with the toe first.

-Page 6 (ii) Perceived ability (stepping-down protocol):

The actual physical activity is relative to leg length (Equation 2.1). Why is it not the case for the perceived ability measured in the stepping-down protocol? If I missed it, perfect then. If it not the case, I think this should be the case because the actual difficulty of stepping down 15 cm is likely dependent on leg length too. In the revision, please provide an equation of this variable similar to the one reported for the actual physical ability.

-Page 6 (e) Data collection and analysis: "A nominal logistic regression model was fitted to the landing-strategy data of each participant. This resulted in a sigmoid model that described the landing strategy as a function of height difference. The height at which the probability of a toe landing equaled the probability of a heel landing was defined as h_{crit} ."

Clean approach.

-Page 9 (Table 1): "Descriptives (N= 61 older adults)"

I would suggest to provide the descriptive data of the participants who were included in the models. As the objective is to predict future falls, my understanding was that the 6 participants who did not complete the 10-month follow-up were not included in the models, therefore resulting in N=55. The table is not consistent in this regard as the percentage of females (61%) is based on n=61, whereas the percentage of fallers (36.4%) is based on n=55.

-Page 9 (d) Exploratory analysis 1: I am wondering whether this subsection is actually an analysis or rather part of the descriptive results. There are no statistics in this subsection, which prevents any generalization of these results to a population larger than the one that was tested in this study. I let the authors decide where this subsection fits best.

-Page 11 (Discussion, first paragraph): "the addition of a misjudgment term, expressed as the interaction between perceived and actual ability did not lead to better prediction of prospective falls than models without a misjudgment term."

One may disagree with this statement as, if I read Table 3 correctly, model 3 (with interaction) fitted the data more accurately than model 2 (without interaction).

Typos/suggestions

-Page 2 (Abstract): "[...] to a default null model, which composed of the conventional measures [...]" => which was composed

-Page 2 (Abstract) : "[...] area under the receiver operating characteristic curve (ROC) [inferior or equal to] 0.64)." => 0.65

-Page 7 (e) Data collection and analysis: "Conflicting categorisations were rated ones more by the first author." => once more

-Page 7 (f) Statistical procedures: "[...] were constructed (eq. 2.2, 2.3, 2.4, 2.5), consisting of both actual physical ability (x_{act}) and perceived ability (i.e., h_{crit})." => including actual physical (x_{act}) and perceived ability terms (i.e., h_{crit})

-Page 7 (f) Statistical procedures: “The four models were compared with a default null model, [...]” => Experimental models (2.3, 2.4, 2.5) were compared to a default null model (2.2) [...]

-Page 9 (Table 1 caption): I know this is obvious, but the authors may consider indicating the meaning of M(SD) and Mdn[IQR] in the caption.

-Page 9 (Table 1):

The brackets are missing for SD of Body height => (9.1).

If there is a decimal for the percentage of fallers (36.4%), to be consistent, the authors may want to add a decimal for the percentage of females (61.0%), even if the decimal is null, this is still more informative than no decimal.

The unit for xact is (-). The authors may prefer to use “arbitrary variable”, which is often used in this case.

-Page 9 (c) Fall prediction models and contribution of misjudgment: “The model coefficients and their statistics for all models are presented in Table 2.” => [...] their statistics are presented in Table 2.

-Page 9 (c) Fall prediction models and contribution of misjudgment: “The misjudgment terms – the interaction term in the second and third model – [...]” => [...] in the first and third models- [...]

-Page 9 (c) Fall prediction models and contribution of misjudgment: “[...] the Bayes factor for these predictors reaching a value of $BF_{10} = 0.45$ and $BF_{10} = 1.88$ respectively.” => [...], respectively.

-Page 9 (c) Fall prediction models and contribution of misjudgment: “However, the coefficients’ BF_{10} was lower than 1, indicating that FESi and QuickScreen only increased the model’s complexity, while their contribution was limited. The BF_{10} values for the experimental models (Table 3) containing our stepping-down measures were lower than 1, showing that the data support the null model over the other experimental models.” It may be worth the exact values of the BF_{10} in brackets to help the reader matching the information in the text with the information in the table.

-Page 11 (Discussions, second paragraph): “Hence, we can conclude that if there is an association of misjudgment with falls, this will be weak.” => [...], this association is weak.

-Page 12 (Discussion):

“These measures were used to reflect the participant’s perceived and actual ability, [...]” => [...] perceived and actual physical ability, [...]

-Page 12 (Discussion): “A more direct measure would give a better indication of when a person truly exceeds his or her actual ability.” => Measures that are more directly related would [...]

-Page 12 (Discussion): “[...] to quantify the misjudgment we determined the actual ability [...] => [...] misjudgment, we [...]

-Page 12 (Discussion): “However, this approach can only be applied in fit older adults and would not be feasible in the more frail individuals.” => [...] in frailer older adults.

-Page 12 (Discussion): “To our knowledge, there is no test available that directly links the perceived and actual ability, without priming or biasing the participant on his/her behaviour.” => [...] actual physical ability, [...]

-Page 13 (Conclusion): "Our findings showed that a misjudgment term, being either an interaction between conventional measures or behavioural measures of perceived and actual physical ability does not add to the prediction of future falls in older adults." => Our findings showed that a misjudgment term does not improve the prediction of future falls in older adults, be it an interaction term between conventional measures or between behavioural measures of perceived and actual physical ability.

-Page 13 (Conclusion): "NK build the setup, [...]" => built

-Page 13 (Conclusion): "[...] and helped shape the research, [...]" => shaping

-References: Please recheck formatting (e.g., some titles capitalize each word, whereas other do not; references to PLoS One article e.g., "PLoS One 12, e0176561-."; some articles show full page numbers whereas over do not (343-345 vs. 343-5); etc.)

Thank you for the opportunity to make comments on your work. I hope it helps.

Matthieu Boisgontier

Decision letter (RSOS-190786.R0)

17-Jul-2019

Dear Mr Kluft,

The editors assigned to your paper ("Does misjudgment in a stepping down paradigm predict falls in an older population?") have now received comments from reviewers. We would like you to revise your paper in accordance with the referee and Associate Editor suggestions which can be found below (not including confidential reports to the Editor). Please note this decision does not guarantee eventual acceptance.

Please submit a copy of your revised paper before 09-Aug-2019. Please note that the revision deadline will expire at 00.00am on this date. If we do not hear from you within this time then it will be assumed that the paper has been withdrawn. In exceptional circumstances, extensions may be possible if agreed with the Editorial Office in advance. We do not allow multiple rounds of revision so we urge you to make every effort to fully address all of the comments at this stage. If deemed necessary by the Editors, your manuscript will be sent back to one or more of the original reviewers for assessment. If the original reviewers are not available, we may invite new reviewers.

When submitting your revised manuscript, you must respond to the comments made by the referees and upload a file "Response to Referees" in "Section 6 - File Upload". Please use this to document how you have responded to the comments, and the adjustments you have made. In

order to expedite the processing of the revised manuscript, please be as specific as possible in your response.

- Data accessibility

If you wish to submit your supporting data or code to Dryad (<http://datadryad.org/>), or modify your current submission to dryad, please use the following link:
<http://datadryad.org/submit?journalID=RSOS&manu=RSOS-190786>

- Competing interests

- Authors' contributions

- Acknowledgements

- Funding statement

Kind regards,

on behalf of Dr Manoj Srinivasan (Associate Editor) and Kevin Padian (Subject Editor)
 openscience@royalsociety.org

Associate Editor's comments (Dr Manoj Srinivasan):

Some minor additional suggestions below in addition to those by the reviewers:

Please also add another paragraph or two providing an explanation/intuition for your Figure 3, specifically, the Receiver Operator Characteristics curve. e.g., explain what the words 'specificity' and 'sensitivity' are computed (their only mention seems to be in the figure) and how the curves relating them for the different models are to be interpreted, as these issues may not be familiar to your intended audience.

Consider a direct visualization of the logistic regression fits, if appropriate: for instance, a scatted plot shown with the exponent in the logistic expression is plotted on the x axis and the fall vs no fall is on the y axis (or whatever else is appropriate)?

Finally, the equations for the probabilities P_{null} , P_1 , etc. are by themselves. Instead, please make them part of complete english sentences, where the variables in the equations are explicitly mentioned. e.g., for the first such equation, something like:

"The probability for the null model (P_{null}) blah blah blah is given the equation:
 <math equation>.

Here and in the the rest of the article, β_i are constant coefficients to be obtained by fitting
 <what data> to this logistic equation."

Please edit to whatever you think is appropriate!

Reviewers' Comments to Author:

Reviewer: 1

Comments to the Author(s)

This study examined whether adding a misjudgement term improved prediction of future falls compared with a) conventional measures of actual (physical measures) and perceived abilities (questionnaires), and b) a novel stepping down paradigm that could quantify behavioural misjudgement. The sample comprised 55 older adults of which 20 reported 1+ falls over a 10 month follow-up period. The main findings were that a default model comprising the conventional measures (FESI and QuickScreen) fitted the data most accurately and that the

inclusion of misjudgement terms did not improve the prediction of future falls in older adults. It was also found the accuracy of all models was low (area under the receiver operating characteristic curve (ROC) ≤ 0.64).

This is an interesting study that uses novel approaches to provide greater understanding of fall risk in older people. The complementary use of both conventional and behavioural measures of fall risk and misjudgement is a major strength of the paper. I have the following issues for consideration.

It is indicated in the methods that the exact measure of physical ability was derived from previous research. Was this measure validated with respect to falls in this larger study?

The combination of the small sample for a fall risk study, a relatively healthy sample and the choice of QuickScreen may have limited the study to detect differences between the faller groups. As indicated in the discussion, real world falls is a "noisy" outcome measure and larger samples ($N > 100$) are usually used to partially address this issue. Studies often focus on multiple falls as this group appear to have chronic physical impairments, whereas those who suffer an occasional fall often perform as well as non-fallers in physical assessments. Indeed, ROC characteristics of the QuickScreen tool (as used here) were based on an outcome of multiple (2+) fallers. The small QuickScreen IQR (2) also indicated little fall risk variation in the sample. Given the small sample, it would be problematic to conduct further analyses, but it would be informative to provide the number of participants who fell one time only and to discuss these issues in a limitations section of the discussion.

Please provide an additional panel to figure 3 depicting falls per group in participants categorised by fall risk matches and mismatches as derived from the conventional measures (FESi and QuickScreen).

It would also be instructive to see correlations between the QuickScreen, FESi, exact and hcrit measures.

Reviewer: 2

Comments to the Author(s)

Title: Does misjudgment in a stepping down paradigm predict falls in an older population?

Journal: Royal Society Open Science

Manuscript ID: RSOS-190786

Date: July 1, 2019

This manuscript reports a single study designed to test whether misjudged physical abilities contribute to predict falls beyond and above actual and perceived physical abilities. Actual abilities were assessed using the weighted average of maximal step length and maximum performance to step over an obstacle. Perceived abilities were assessed based on a nominal logistic regression model that fitted the toe versus heel landing-strategy data in a 6-height stepping-down protocol. Misjudged abilities were modeled by the interaction term between the actual and perceived abilities. The data from 55 older adults (20 fallers) were included in Bayesian logistic-regression models. Results showed that the interaction terms improved the fit of the model but that the null model was the most accurate one. All models showed low accuracy with areas under the ROC curves < 0.66 .

General comments

This manuscript is very well written, it was a pleasure to read it. The introduction provides an accurate and complete overview of the existing literature related to falls and misjudged physical abilities. The approach used to address the research question is clever. The statistics used are appropriate and go beyond the usual standards in our field. The results do not support the

hypothesis but provide useful and clear information that will serve the implementation of future study designs.

Specific comments

-Page 5 (Objectives): “We predict that overestimation is associated with prospective falls, and thus a combination of high perceived and low actual ability should be associated with higher fall incidence. Hence, an interaction between perceived ability (i.e., h_{crit}) and actual ability, as suggested by [12], would improve the power to predict fall risk compared to a model based on merely physical and falls efficacy measures.”

Clever approach to the research question.

-Page 5 (a) Participants: “Participants were included when they had a mini-mental state examination score above 18 points”

It may be worth making the meaning of this 18-point threshold explicit to the reader (e.g. no to mild cognitive impairment).

-Page 5 (c) Falls: “Falls were monitored for 10 months using telephone calls”

If the frequency (even a rough one) of these phone calls could be reported, this may be useful information to readers.

-Page 6 (ii) Perceived ability (stepping-down protocol):

In future studies, as this task is central to the research question, it may be a good idea to control for passive and/or active ankle range of motion, especially dorsiflexion. Limited range of dorsiflexion forces the heel of the upper foot to leave the step earlier thereby reducing stability and increasing the likelihood that the participants try to contact the lower step as fast as possible with the lower foot, which would be done by contacting the step with the toe first.

-Page 6 (ii) Perceived ability (stepping-down protocol):

The actual physical activity is relative to leg length (Equation 2.1). Why is it not the case for the perceived ability measured in the stepping-down protocol? If I missed it, perfect then. If it not the case, I think this should be the case because the actual difficulty of stepping down 15 cm is likely dependent on leg length too. In the revision, please provide an equation of this variable similar to the one reported for the actual physical ability.

-Page 6 (e) Data collection and analysis: “A nominal logistic regression model was fitted to the landing-strategy data of each participant. This resulted in a sigmoid model that described the landing strategy as a function of height difference. The height at which the probability of a toe landing equaled the probability of a heel landing was defined as h_{crit} .”

Clean approach.

-Page 9 (Table 1): “Descriptives (N= 61 older adults)”

I would suggest to provide the descriptive data of the participants who were included in the models. As the objective is to predict future falls, my understanding was that the 6 participants who did not complete the 10-month follow-up were not included in the models, therefore resulting in N=55. The table is not consistent in this regard as the percentage of females (61%) is based on n=61, whereas the percentage of fallers (36.4%) is based on n=55.

-Page 9 (d) Exploratory analysis 1: I am wondering whether this subsection is actually an analysis or rather part of the descriptive results. There are no statistics in this subsection, which prevents any generalization of these results to a population larger than the one that was tested in this study. I let the authors decide where this subsection fits best.

-Page 11 (Discussion, first paragraph): “the addition of a misjudgment term, expressed as the

interaction between perceived and actual ability did not lead to better prediction of prospective falls than models without a misjudgment term.”

One may disagree with this statement as, if I read Table 3 correctly, model 3 (with interaction) fitted the data more accurately than model 2 (without interaction).

Typos/suggestions

-Page 2 (Abstract): “[...] to a default null model, which composed of the conventional measures [...]” => which was composed

-Page 2 (Abstract) : “[...] area under the receiver operating characteristic curve (ROC) [inferior or equal to] 0.64.” => 0.65

-Page 7 (e) Data collection and analysis: “Conflicting categorisations were rated ones more by the first author.” => once more

-Page 7 (f) Statistical procedures: “[...] were constructed (eq. 2.2, 2.3, 2.4, 2.5), consisting of both actual physical ability (xact) and perceived ability (i.e., hcrit).” => including actual physical (xact) and perceived ability terms (i.e., hcrit)

-Page 7 (f) Statistical procedures: “The four models were compared with a default null model, [...]” => Experimental models (2.3, 2.4, 2.5) were compared to a default null model (2.2) [...]

-Page 9 (Table 1 caption): I know this is obvious, but the authors may consider indicating the meaning of M(SD) and Mdn[IQR] in the caption.

-Page 9 (Table 1):

The brackets are missing for SD of Body height => (9.1).

If there is a decimal for the percentage of fallers (36.4%), to be consistent, the authors may want to add a decimal for the percentage of females (61.0%), even if the decimal is null, this is still more informative than no decimal.

The unit for xact is (-). The authors may prefer to use “arbitrary variable”, which is often used in this case.

-Page 9 (c) Fall prediction models and contribution of misjudgment: “The model coefficients and their statistics for all models are presented in Table 2.” => [...] their statistics are presented in Table 2.

-Page 9 (c) Fall prediction models and contribution of misjudgment: “The misjudgment terms – the interaction term in the second and third model – [...]” => [...] in the first and third models- [...]

-Page 9 (c) Fall prediction models and contribution of misjudgment: “[...] the Bayes factor for these predictors reaching a value of $BF_{10} = 0.45$ and $BF_{10} = 1.88$ respectively.” => [...], respectively.

-Page 9 (c) Fall prediction models and contribution of misjudgment: “However, the coefficients’ BF_{10} was lower than 1, indicating that FESi and QuickScreen only increased the model’s complexity, while their contribution was limited. The BF_{10} values for the experimental models (Table 3) containing our stepping-down measures were lower than 1, showing that the data support the null model over the other experimental models.” It may be worth the exact values of the BF_{10} in brackets to help the reader matching the information in the text with the information in the table.

-Page 11 (Discussions, second paragraph): “Hence, we can conclude that if there is an association of misjudgment with falls, this will be weak.” => [...], this association is weak.

-Page 12 (Discussion):

“These measures were used to reflect the participant’s perceived and actual ability, [...]” => [...] perceived and actual physical ability, [...]

-Page 12 (Discussion): “A more direct measure would give a better indication of when a person truly exceeds his or her actual ability.” => Measures that are more directly related would [...]

-Page 12 (Discussion): “[...] to quantify the misjudgment we determined the actual ability [...] => [...] misjudgment, we [...]

-Page 12 (Discussion): “However, this approach can only be applied in fit older adults and would not be feasible in the more frail individuals.” => [...] in frailer older adults.

-Page 12 (Discussion): “To our knowledge, there is no test available that directly links the perceived and actual ability, without priming or biasing the participant on his/her behaviour.” => [...] actual physical ability, [...]

-Page 13 (Conclusion): “Our findings showed that a misjudgment term, being either an interaction between conventional measures or behavioural measures of perceived and actual physical ability does not add to the prediction of future falls in older adults.” => Our findings showed that a misjudgment term does not improve the prediction of future falls in older adults, be it an interaction term between conventional measures or between behavioural measures of perceived and actual physical ability.

-Page 13 (Conclusion): “NK build the setup, [...]” => built

-Page 13 (Conclusion): “[...] and helped shape the research, [...]” => shaping

-References: Please recheck formatting (e.g., some titles capitalize each word, whereas other do not; references to PLoS One article e.g., “PLoS One 12, e0176561-.”; some articles show full page numbers whereas over do not (343-345 vs. 343-5); etc.)

Thank you for the opportunity to make comments on your work. I hope it helps.

Matthieu Boisgontier

Author's Response to Decision Letter for (RSOS-190786.R0)

See Appendix A.

RSOS-190786.R1 (Revision)

Review form: Reviewer 1

Is the manuscript scientifically sound in its present form?

Yes

Are the interpretations and conclusions justified by the results?

Yes

Is the language acceptable?

Yes

Do you have any ethical concerns with this paper?

No

Have you any concerns about statistical analyses in this paper?

No

Recommendation?

Accept as is

Comments to the Author(s)

The authors have addressed all my initial issues, and I have no additional issues to raise.

Review form: Reviewer 2 (Matthieu Boisgontier)

Is the manuscript scientifically sound in its present form?

Yes

Are the interpretations and conclusions justified by the results?

Yes

Is the language acceptable?

Yes

Do you have any ethical concerns with this paper?

No

Have you any concerns about statistical analyses in this paper?

No

Recommendation?

Accept as is

Comments to the Author(s)

Thank you for the opportunity to review this paper.

MB

Decision letter (RSOS-190786.R1)

08-Oct-2019

Dear Mr Klufft,

I am pleased to inform you that your manuscript entitled "Does misjudgment in a stepping down paradigm predict falls in an older population?" is now accepted for publication in Royal Society Open Science.

Now that your manuscript has been accepted for publication, please also ensure that your OSF record is made public; ensuring to update the private "review-only" link in your main manuscript once you receive the proof of your article.

Kind regards,
Lianne Parkhouse
Royal Society Open Science
openscience@royalsociety.org

on behalf of Dr Manoj Srinivasan (Associate Editor) and Professor Kevin Padian (Subject Editor)
openscience@royalsociety.org

Associate Editor Comments to Author (Dr Manoj Srinivasan):

The authors have addressed all reviewer and editorial remarks thoughtfully.

Reviewer comments to Author:

Reviewer: 1
Comments to the Author(s)

The authors have addressed all my initial issues, and I have no additional issues to raise.

Reviewer: 2
Comments to the Author(s)

Thank you for the opportunity to review this paper.
MB

Appendix A

Title: Does misjudgment in a stepping down paradigm predict falls in an older population?

Journal: Royal Society Open Science

Manuscript ID: RSOS-190786

Date: July 22, 2019

Dear Mr Kluff,

The editors assigned to your paper ("Does misjudgment in a stepping down paradigm predict falls in an older population?") have now received comments from reviewers. We would like you to revise your paper in accordance with the referee and Associate Editor suggestions which can be found below (not including confidential reports to the Editor). Please note this decision does not guarantee eventual acceptance.

Please submit a copy of your revised paper before 09-Aug-2019. Please note that the revision deadline will expire at 00.00am on this date. If we do not hear from you within this time then it will be assumed that the paper has been withdrawn. In exceptional circumstances, extensions may be possible if agreed with the Editorial Office in advance. We do not allow multiple rounds of revision so we urge you to make every effort to fully address all of the comments at this stage. If deemed necessary by the Editors, your manuscript will be sent back to one or more of the original reviewers for assessment. If the original reviewers are not available, we may invite new reviewers.

- Data accessibility

<http://datadryad.org/submit?journalID=RSOS&manu=RSOS-190786>

- Competing interests

- Authors' contributions

- Acknowledgements

- Funding statement

Kind regards,

Lianne Parkhouse

Editorial Coordinator

on behalf of Dr Manoj Srinivasan (Associate Editor) and Kevin Padian (Subject Editor)

Associate Editor's comments (Dr Manoj Srinivasan):

Some minor additional suggestions below in addition to those by the reviewers:

Editor's suggestion 1:

Please also add another paragraph or two providing an explanation/intuition for your Figure 3, specifically, the Receiver Operator Characteristics curve. e.g., explain what the words 'specificity' and 'sensitivity' are computed (their only mention seems to be in the figure) and how the curves relating them for the different models are to be interpreted, as these issues may not be familiar to your intended audience.

We thank the editors for highlighting this. We have added a sentence to the manuscript regarding the sensitivity and specificity. Moreover, we have extended the caption of the figure to clarify these concepts

“To better understand and compare the predictive value of the different models, we computed the receiver operating characteristic (ROC) curvature for each model (see Figure 3). In the ROC curves, the sensitivity (i.e., the proportion of actual positives that are correctly identified) is plotted as a function of the specificity (i.e., the proportion of actual negatives that are correctly identified) for different criterion values (i.e., cut-points) of the model. The area under the ROC curvatures (ROC_{auc}) is a measure of how accurate the model can distinguish between fallers and non-fallers. In our sample....” - Results, Exploratory analysis 2 (page 9).

“Figure 3. Receiver operating characteristic curves for the individual models: sensitivity (the proportion of actual positives that are correctly identified), and specificity (the proportion of actual negatives that are correctly identified), are depicted for different criterion values of the model. The area under the ROC curve (ROC_{auc}) is given as a measure of the overall accuracy of the model, a larger ROC_{auc} indicates better performance of the model to distinguish between fallers and non-fallers. The diagonal line (with an auc of 0.5) represents a non-discriminatory model.” -

Caption Figure 3 (page 10)

Editor's suggestion 2:

Consider a direct visualization of the logistic regression fits, if appropriate: for instance, a scatted plot shown with the exponent in the logistic expression is plotted on the x axis and the fall vs no fall is on the y axis (or whatever else is appropriate)?

We considered this analysis, but as the logistic regression models consisted of more than one determinant, it was not possible to visualise the model's behaviour accurately in a 2D plot (such as a scatter plot). Instead, we have chosen to visualise the results as done so in Figure 2.

Editor's suggestion 3:

Finally, the equations for the probabilities P_{null} , P_1 , etc. are by themselves. Instead, please make them part of complete english sentences, where the variables in the equations are explicitly mentioned. e.g., for the first such equation, something like:

"The probability for the null model (P_{null}) blah blah blah is given the equation:

<math equation>.

Here and in the the rest of the article, β_i are constant coefficients to be obtained by fitting <what data> to this logistic equation."

Please edit to whatever you think is appropriate!

We have edited the sections with equations as the editor suggested, and integrated them in the text:

"The probability of a fall (P_{null}) as a function of the FESi and the QS in the first model is given by the equation:

<math equation>. (2.2) " - Methods, Statistical procedure (page 5)

"In the first experimental model, the probability of a fall (P_1) as a function of the FESi and the QS is given by the equation:

<math equation>. (2.3) " - Methods, First model (page 5)

"In the second experimental model, the probability of a fall (P_2) as a function of x_{act} and h_{crit} is computed by the equation:

<math equation>. (2.4) " - Methods, Second model (page 6)

"The probability of a fall (P_3) in the third model is given by the equation:

<math equation>, (2.5)

in which the interaction $x_{\text{act}} \times h_{\text{crit}}$ represents the misjudgment term. " - Methods, Second model (page 6)

Reviewer 1

This study examined whether adding a misjudgement term improved prediction of future falls compared with a) conventional measures of actual (physical measures) and perceived abilities (questionnaires), and b) a novel stepping down paradigm that could quantify behavioural misjudgement. The sample comprised 55 older adults of which 20 reported 1+ falls over a 10 month follow-up period. The main findings were that a default model comprising the conventional measures (FESI and QuickScreen) fitted the data most accurately and that the inclusion of misjudgement terms did not improve the prediction of future falls in older adults. It was also found the accuracy of all models was low (area under the receiver operating characteristic curve (ROC) ≤ 0.64).

This is an interesting study that uses novel approaches to provide greater understanding of fall risk in older people. The complementary use of both conventional and behavioural measures of fall risk and misjudgement is a major strength of the paper. I have the following issues for consideration.

We thank the reviewer for the constructive feedback and we have incorporated all suggestions in our revised manuscript.

Comment 1

It is indicated in the methods that the exact measure of physical ability was derived from previous research. Was this measure validated with respect to falls in this larger study?

The measure (x_{act}) is composed of two measures that have shown to be reliable and consistent in a larger study from our group (Weijer et al., 2019), but have not yet been validated with respect to future falls. There is no evidence that either measure (i.e., step-over or step-length ability) would better represent the general concept of stepping ability. Therefore, we decided to combine these measures using equation 2.1.

These measures have not yet been validated with respect to falls. Yet, in this study we were interested in measures that could quantify stepping ability, so we could compare this with the perceived stepping ability.

Comment 2

The combination of the small sample for a fall risk study, a relatively healthy sample and the choice of QuickScreen may have limited the study to detect differences between the faller groups. As indicated in the discussion, real world falls is a “noisy” outcome measure and larger samples ($N > 100$) are usually used to partially address this issue. Studies often focus on multiple falls as this group appear to have chronic physical impairments, whereas those who suffer an occasional fall often perform as well as non-fallers in physical assessments. Indeed, ROC characteristics of the QuickScreen tool (as used here) were based on an outcome of multiple (2+) fallers. The small QuickScreen IQR (2) also indicated little fall risk variation in the sample. Given the small sample, it would be problematic to conduct further analyses, but it would be informative to provide the number of participants who fell one time only and to discuss these issues in a limitations section of the discussion.

We agree on this comment and addressed this limitation more specifically in the discussion:

“[...] The median number of risk factors identified by the QuickScreen in our sample was 4, which was higher than what was identified in an external validation study (median number of risk factors = 3 [41]). Furthermore it should be noted that the QuickScreen tool's ROC_{AUC} values were developed in a larger cohort and on the basis of multiple falls data. Recurrent fallers are likely to suffer from more chronic impairments in either perceived or actual ability. Only 14 participants in our study fell more than once during follow up, which was a too small number to distinguish recurrent fallers from occasional fallers and non-fallers.” - Discussion, Paragraph 3 (page 10)

Comment 3

Please provide an additional panel to figure 3 depicting falls per group in participants categorised by fall risk matches and mismatches as derived from the conventional measures (FESi and QuickScreen).

We have provided a panel in figure 2 depicting falls per group in which groups were separated on the basis of FESi and QuickScreen, and have introduced this analysis in the Results, Exploratory analysis 1 (page 8) section.

Comment 4

It would also be instructive to see correlations between the QuickScreen, FESi, x_{act} and h_{crit} measures.

We thank the reviewer for this suggestion, we have added this in the methods as the “input variables and priors” section.

“(iv) Input variables and priors

For all models, each input variable was standardised, and weakly informative priors were assigned to all model predictors [31]. The prior for the model's intercept followed a zero centred Cauchy distribution with scale 10, and a zero centred Cauchy distribution with scale 2.5 was selected for the predictor coefficients priors, thereby following the recommendations of [32].

Collinearity of the predictors was assessed using the variance inflation factor, and showed that there was no severe collinearity that would have affected the analysis ($VIF < 1.8$). Hence, correlations between predictors were low ($r_{(QuickScreen, FESi)} = 0.06$; $r_{(x_{act}, h_{crit})} = 0.20$).” - Methods, Input variables and priors (page 6)

Reviewer 2

This manuscript reports a single study designed to test whether misjudged physical abilities contribute to predict falls beyond and above actual and perceived physical abilities. Actual abilities

were assessed using the weighted average of maximal step length and maximum performance to step over an obstacle. Perceived abilities were assessed based on a nominal logistic regression model that fitted the toe versus heel landing-strategy data in a 6-height stepping-down protocol. Misjudged abilities were modeled by the interaction term between the actual and perceived abilities. The data from 55 older adults (20 fallers) were included in Bayesian logistic-regression models. Results showed that the interaction terms improved the fit of the model but that the null model was the most accurate one. All models showed low accuracy with areas under the ROC curves $<.66$).

General comments

This manuscript is very well written, it was a pleasure to read it. The introduction provides an accurate and complete overview of the existing literature related to falls and misjudged physical abilities. The approach used to address the research question is clever. The statistics used are appropriate and go beyond the usual standards in our field. The results do not support the hypothesis but provide useful and clear information that will serve the implementation of future study designs.

We thank the reviewer for the positive and extensive feedback. We revised our manuscript to address the reviewer's suggestions and to improve the manuscript.

Specific comment 1

-Page 5 (Objectives): "We predict that overestimation is associated with prospective falls, and thus a combination of high perceived and low actual ability should be associated with higher fall incidence. Hence, an interaction between perceived ability (i.e., h_{crit}) and actual ability, as suggested by [12], would improve the power to predict fall risk compared to a model based on merely physical and falls efficacy measures."

Clever approach to the research question.

We thank the reviewer for the compliment.

Specific comment 2

-Page 5 (a) Participants: "Participants were included when they had a mini-mental state examination score above 18 points"

It may be worth making the meaning of this 18-point threshold explicit to the reader (e.g. no to mild cognitive impairment).

We agree and added this information in the description of the participants:

"Sixty-two older adults (age ≥ 65 years, median age 73.5 [IQR 10] years, 37 females, 20 fallers) participated in the study. Participants were included when they had a mini-mental state examination score above 18 points (i.e., no to mild cognitive decline, excluding severe cognitive impairment), were able to walk 20 meters continuously without becoming short of breath, experiencing dizziness, or perceiving pain in - or pressure on the chest." - Methods, Participants (page 3)

Specific comment 3

-Page 5 (c) Falls: “Falls were monitored for 10 months using telephone calls”

If the frequency (even a rough one) of these phone calls could be reported, this may be useful information to readers.

The participants were called once a month, for 10 consecutive months. We have added the following text to the manuscript:

“Falls were monitored for 10 months using fall diaries and monthly telephone calls to ask about the occurrence of falls over the previous month.” - Methods, Falls (page 3)

Specific comment 4

-Page 6 (ii) Perceived ability (stepping-down protocol):

In future studies, as this task is central to the research question, it may be a good idea to control for passive and/or active ankle range of motion, especially dorsiflexion. Limited range of dorsiflexion forces the heel of the upper foot to leave the step earlier thereby reducing stability and increasing the likelihood that the participants try to contact the lower step as fast as possible with the lower foot, which would be done by contacting the step with the toe first.

We thank the reviewer for bringing this potentially confounding effect to our attention. The ankle range of motion would indeed be an interesting factor to incorporate in other studies in which we use this or similar stepping-down paradigms. We have added a new section to the discussion of the manuscript.

“Participant’s perceived ability was assessed using a stepping-down paradigm. The strength of such a paradigm is that the perceived ability is not explicitly assessed. However, the stepping-down behaviour might have been affected by the participant’s anthropometry. For instance, limited dorsiflexion range-of-motion could induce toe-off earlier in the gait cycle, which may affect one’s stability. This physical limitation may drive participants to select a toe-landing strategy where a heel landing would have been more appropriate. Hence, we recommend to take passive/active ankle range of motion into account in future research that implements a stepping-down paradigm.” - Discussion, 5th paragraph (page 11)

Specific comment 5

-Page 6 (ii) Perceived ability (stepping-down protocol):

The actual physical activity is relative to leg length (Equation 2.1). Why is it not the case for the perceived ability measured in the stepping-down protocol? If I missed it, perfect then. If it not the case, I think this should be the case because the actual difficulty of stepping down 15 cm is likely dependent on leg length too. In the revision, please provide an equation of this variable similar to the one reported for the actual physical ability.

The reviewer is correct that we did not relate the perceived ability measure to leg length, and that this would be most appropriate to do so. Hence, text, tables and figures have been adjusted after

normalising the perceived ability by leg length. It did not affect our results. We clarified the normalisation in the text.

“A nominal logistic regression model was fitted to the landing-strategy data of each participant. This resulted in a sigmoid model that described the landing strategy as a function of height difference. The height at which the probability of a toe landing (P_{toe}) equaled the probability of a heel landing (P_{heel}) was defined as h_{crit} [23] normalised to participant's leg length.” - Methods, Data collection and analysis (page 5)

Specific comment 6

-Page 6 (e) Data collection and analysis: “A nominal logistic regression model was fitted to the landing-strategy data of each participant. This resulted in a sigmoid model that described the landing strategy as a function of height difference. The height at which the probability of a toe landing equaled the probability of a heel landing was defined as h_{crit} .”

Clean approach.

We thank the reviewer for the compliment.

Specific comment 7

-Page 9 (Table 1): “Descriptives (N= 61 older adults)”

I would suggest to provide the descriptive data of the participants who were included in the models. As the objective is to predict future falls, my understanding was that the 6 participants who did not complete the 10-month follow-up were not included in the models, therefore resulting in N=55. The table is not consistent in this regard as the percentage of females (61%) is based on n=61, whereas the percentage of fallers (36.4%) is based on n=55.

We thank the reviewer for highlighting this inconsistency. We have changed the table accordingly, which now overall describes the data of n=55.

Specific comment 8

-Page 9 (d) Exploratory analysis 1: I am wondering whether this subsection is actually an analysis or rather part of the descriptive results. There are no statistics in this subsection, which prevents any generalization of these results to a population larger than the one that was tested in this study. I let the authors decide where this subsection fits best.

We have decided to keep the structure as it was. We did not plan to do this analysis and there have been a few arbitrary choices in this analysis that we would feel unhappy to represent as if planned.

Specific comment 9

-Page 11 (Discussion, first paragraph): “the addition of a misjudgment term, expressed as the interaction between perceived and actual ability did not lead to better prediction of prospective falls than models without a misjudgment term.”

One may disagree with this statement as, if I read Table 3 correctly, model 3 (with interaction) fitted the data more accurately than model 2 (without interaction).

We have rewritten this section to be more precise about the results.

“None of the experimental models performed better than the null model that contained solely a clinical measure and questionnaire (i.e., FESi and QuickScreen). Adding a misjudgment term to the null model did not improve the predictive quality of the model (e.g., null model’s LOO = 74.52, versus 1st model’s LOO = 76.14). Moreover, we found that the addition of a misjudgment term, led to only very minor improvement in prediction of prospective falls when expressed as the interaction between perceived and actual ability of behavioural measures (i.e., xact x hcrit) (e.g., 3rd model’s LOO = 75.39, versus 2nd model’s LOO = 77.40).” - Discussion, first paragraph (page 9)

Typos/suggestions

The authors appreciate the reviewer’s suggestions and that he highlighted the typos in the manuscript. We have addressed all typos/suggestions in the revised document.

-Page 2 (Abstract): “[...] to a default null model, which composed of the conventional measures [...]” => which was composed

-Page 2 (Abstract): “[...] area under the receiver operating characteristic curve (ROC) [inferior or equal to] 0.64).” => 0.65

-Page 7 (e) Data collection and analysis: “Conflicting categorisations were rated ones more by the first author.” => once more

-Page 7 (f) Statistical procedures: “[...] were constructed (eq. 2.2, 2.3, 2.4, 2.5), consisting of both actual physical ability (xact) and perceived ability (i.e., hcrit).” => including actual physical (xact) and perceived ability terms (i.e., hcrit)

-Page 7 (f) Statistical procedures: “The four models were compared with a default null model, [...]” => Experimental models (2.3, 2.4, 2.5) were compared to a default null model (2.2) [...]

-Page 9 (Table 1 caption): I know this is obvious, but the authors may consider indicating the meaning of M(SD) and Mdn[IQR] in the caption.

-Page 9 (Table 1):

The brackets are missing for SD of Body height => (9.1).

If there is a decimal for the percentage of fallers (36.4%), to be consistent, the authors may want to add a decimal for the percentage of females (61.0%), even if the decimal is null, this is still more informative than no decimal.

The unit for xact is (-). The authors may prefer to use “arbitrary variable”, which is often used in this case.

-Page 9 (c) Fall prediction models and contribution of misjudgment: “The model coefficients and their statistics for all models are presented in Table 2.” => [...] their statistics are presented in Table 2.

-Page 9 (c) Fall prediction models and contribution of misjudgment: “The misjudgment terms – the interaction term in the second and third model – [...]” => [...] in the first and third models– [...]

-Page 9 (c) Fall prediction models and contribution of misjudgment: “[...] the Bayes factor for these predictors reaching a value of $BF_{10} = 0.45$ and $BF_{10} = 1.88$ respectively.” => [...], respectively.

-Page 9 (c) Fall prediction models and contribution of misjudgment: “However, the coefficients’ BF_{10} was lower than 1, indicating that FESi and QuickScreen only increased the model’s complexity, while their contribution was limited. The BF_{10} values for the experimental models (Table 3) containing our stepping-down measures were lower than 1, showing that the data support the null model over the other experimental models.” It may be worth the exact values of the BF_{10} in brackets to help the reader matching the information in the text with the information in the table.

-Page 11 (Discussions, second paragraph): “Hence, we can conclude that if there is an association of misjudgment with falls, this will be weak.” => [...], this association is weak.

-Page 12 (Discussion): “These measures were used to reflect the participant’s perceived and actual ability, [...]” => [...] perceived and actual physical ability, [...]

-Page 12 (Discussion): “A more direct measure would give a better indication of when a person truly exceeds his or her actual ability.” => Measures that are more directly related would [...]

-Page 12 (Discussion): “[...] to quantify the misjudgment we determined the actual ability [...] => [...] misjudgment, we [...]

-Page 12 (Discussion): “However, this approach can only be applied in fit older adults and would not be feasible in the more frail individuals.” => [...] in frailer older adults.

-Page 12 (Discussion): “To our knowledge, there is no test available that directly links the perceived and actual ability, without priming or biasing the participant on his/her behaviour.” => [...] actual physical ability, [...]

-Page 13 (Conclusion): “Our findings showed that a misjudgment term, being either an interaction between conventional measures or behavioural measures of perceived and actual physical ability does not add to the prediction of future falls in older adults.” => Our findings showed that a misjudgment term does not improve the prediction of future falls in older adults, be it an interaction term between conventional measures or between behavioural measures of perceived and actual physical ability.

-Page 13 (Conclusion): “NK build the setup, [...]” => built

-Page 13 (Conclusion): “[...] and helped shape the research, [...]” => shaping

-References: Please recheck formatting (e.g., some titles capitalize each word, whereas other do not; references to PLoS One article e.g., “PLoS One 12, e0176561–.”; some articles show full page numbers whereas over do not (343-345 vs. 343–5); etc.)

Thank you for the opportunity to make comments on your work. I hope it helps.

Matthieu Boisgontier